

# Is there a hybridization barrier between *Gentiana lutea* color morphs?

María Losada[1], Tania Veiga[2], Javier Guitián[2], José Guitián[1],
Pablo Guitián[2] and Mar Sobral[1]

[1] Department of Cell Biology and Ecology, Area of Ecology, Biology School, Universidad de Santiago de Compostela, Santiago de Compostela, A Coruña, Spain
[2] Department of Botany, Biology School, Universidad de Santiago de Compostela, Santiago de Compostela, A Coruña, Spain

## ABSTRACT

In *Gentiana lutea* two varieties are described: *G. lutea* var. *aurantiaca* with orange corolla colors and *G. lutea* var. *lutea* with yellow corolla colors. Both color varieties co-occur in NW Spain, and pollinators select flower color in this species. It is not known whether a hybridization barrier exists between these *G. lutea* color varieties. We aim to test the compatibility between flower color varieties in *G. lutea* and its dependence on pollen vectors. Within a sympatric population containing both flower color morphs, we analyzed differences in reproductive success (number, weight, viability and germinability of seeds) depending on fertilization treatments (autogamy and xenogamy within variety and among varieties). We found a 93% reduction in number of seeds and a 37% reduction in seed weight respectively of autogamy treatments compared to xenogamy crossings. Additionally, reproductive success is higher within color varieties than among varieties, due to a 45% seed viability reduction on hybrids from different varieties. Our results show that *G. lutea* reproductive success is strongly dependent on pollinators and that a partial hybridization barrier exists between *G. lutea* varieties.

Corresponding author
Mar Sobral,
sobral.bernal.mar@gmail.com

## INTRODUCTION

Pollen vectors can drive plant evolution and diversification throughout their selection on floral traits (*Darwin, 1859*; *Darwin, 1862*; *Thompson, 1994*; *Barrett & Harder, 1996*; *Charlesworth, 2006*). Most angiosperms need vectors for pollen transfer between plants, which are mainly insects but can also be other animals and to a lesser extent wind or water (*Harder & Barrett, 1996*; *Ackerman, 2000*; *Ollerton, Winfree & Tarrant, 2011*). Different floral strategies were developed to attract these animals (*Ghazoul, 2006*), which may affect plant fitness to a large degree (*Waser, 1983*; *Conner & Rush, 1996*). This plant–pollinator relationship promotes evolution of species involved in such an interaction. Thus, the degree of dependence on animal pollinators might affect the strength of selection and therefore the likelihood of species diversification.

*Gentiana lutea* L. shows flower color variation from orange to yellow both, within and among populations (*Sobral et al., 2015*) at the western extreme of the distribution range. Two different varieties are described for *Gentiana lutea* L. depending on flower color: *Gentiana lutea* L. var. *aurantiaca* (M. Laínz) showing orange corolla colors and *Gentiana lutea* L. var. *lutea* showing yellow corollas (*Laínz, 1982*; *Renobales, 2003*). *Gentiana lutea*'s flower color is a trait with a genetic basis (*Zhu et al., 2002*; *Zhu et al., 2003*) and there are genetic differences among populations (*González-López et al., 2014*). We know that flower color variation among populations in this species across NW Spain, is not related to abiotic environmental factors such as elevation, temperature, radiation and rainfall (*Veiga et al., 2015a*). Additionally, we know that pollinators exert selective pressures on *G. lutea* flower color (*Veiga et al., 2015b*) and play a role in flower color differentiation among *G. lutea* populations (*Sobral et al., 2015*). *Rossi (2012)* described *G. lutea* as a partial self-compatible species. However, others cite *G. lutea* as a self-incompatible species, which may need animal pollinators to reproduce (*Hegi, 1927*; *Kèry, Matthies & Spillmann, 2000*; *Kozuharova & Anchev, 2006*; *González-López et al., 2014*).

The aim of this study is to test if there is some degree of incompatibility between *Gentiana lutea* color varieties due to a partial hybridization barrier. If a hybridization barrier exists between *G. lutea* color morphs, we might expect crossings among varieties to have a lower reproductive success (*Hauser, Jørgensen & Østergård, 1998*; *Hauser, Shaw & Østergård, 1998*). Additionally, if *G. lutea* color morphs result from selective pressures exerted by pollinators (*Veiga et al., 2015b*; *Sobral et al., 2015*), we would expect *G. lutea* to strongly depend on the pollinating vectors exerting this selection. Thus, *G. lutea* crossings among different individuals would have higher reproductive success than fertilizations within-individual plants. In order to investigate our hypothesis, we proposed the following questions; (i) Are there any differences in reproductive success (seed number, seed weight, seed viability and seed germinability) between within-color variety crossings and among-color variety crossings? (ii) Are there any differences in reproductive success between self-pollinated seeds and seeds coming from among individual crossings?

## MATERIALS & METHODS

*Gentiana lutea* L. (Gentianaceae) is a herbaceous perennial plant distributed along the Central and Southern European mountains typically growing on livestock grazing grasslands and hillsides from montane to sub-alpine habitats, approximately from 800 to 2.500 m a.s.l. (*Hesse, Rees & Müller-Schärer, 2007*; *Anchisi et al., 2010*). This long-lived species presents a rhizome, which develops one (rarely two or three) unbranched stout stem with basal leaves, up to 200 cm tall (*Renobales, 2012*). Flowering occurs in summer (June–July) when the fertile stems show flowers, grouped in pseudo-whorls, which bloom spirally; and the inflorescence develops in succession from apex to base (*Kozuharova, 1994*). There are two different varieties that differ in flower color: *Gentiana lutea* L. var. *aurantiaca* (M. Laínz) with orange flowers and *Gentiana lutea* var. *lutea* L. with yellow flowers (*Laínz, 1982*; *Renobales, 2003*). *Gentiana lutea* flowers show rotate corollas with lobes (*Renobales, 2012*), which facilitate the access of pollinators to the nectaries and are visited by at least 11
families of insects, belonging to four orders (*Rossi et al., 2014*). The main flower visitors of *G. lutea* plants at the Cantabrian Mountains are bumblebees, followed by cuckoo bumblebees and honeybees (*Sobral et al., 2015*). Fruits hold many flattened, elliptic and winged seeds disseminated through anemochory (*Struwe & Albert, 2002*).

We carried out experimental manipulations at one *G. lutea* population in León, Spain (43°03′N; 6°04′W; 1,600 m a.s.l.) in July 2012. For field experiments, we received field permit from Environmental Territorial Service from León, Territorial Delegation of Government of Spain, Regional Government of Castilla and León (Identifier:12_LE_325_RNA_PuebladeLilio_INV; Reference: 06.01.013.016/ROT/abp; File number: AEN/LE/103/12). Note that in this study, we will distinguish two different flower color classes (orange and yellow). Both varieties do not differ in their UV reflectance but they do differ in their visible reflectance (see *Veiga et al., 2015b*); thus, we use visible reflectance to identify color varieties. We haphazardly chose five flowers on each of 25 orange-flowered plants and 25 yellow-flowered plants. Flower buds, except the control group, were bagged with tulle before flower opening and until fruit formation in order to avoid contact with pollen vectors.

We created the following treatments: (1) Control group (C), in which we applied no treatment, so natural pollination occurred; (2) Spontaneous autogamy (Sa), in which pollen was not applied manually and only pollen from the same flower may spontaneously arrive to the ovary; (3) Facilitated autogamy (Fa), in which we applied pollen manually from the flower's own anthers; (4) Facilitated xenogamy within varieties (Fxw), in which we emasculated flowers by cutting the stamens before pollen release in order to prevent the entry of flower's own pollen, and applied pollen manually from other plants of the same flower color; (5) Facilitated xenogamy among varieties (Fxa), in which we emasculated the flowers and applied pollen manually from plants with different flower color.

Reproductive success can be quantified as the number of fertile descendants produced by an individual throughout its life. It is not feasible to quantify reproductive success in long-lived species in these terms, so seed production is considered a good estimate (see the review of *Kingsolver et al., 2001*). Other measures of reproductive success are proportion of viable seeds and seed germinability. We used four measures of reproductive success: number of seeds, weight of seeds (mg), proportion of viable seeds (seed viability) and proportion of germinated seeds (seed germinability). Due to manipulations some bags were opened, thus plants with the five treatments were down to 26 (18 orange and 8 yellow-flowering individuals). One hundred and thirty ripped fruits were collected before opening, being careful to ensure that the seeds were fully formed. On each fruit we measured seed weight (mg) and counted the number of seeds and the number of ovules not developed; from the sum of non-fertilized ovules and the seeds, we obtained the total number of ovules. Total number of ovules did not vary among treatments (please see Supplemental Information 2), thus we used the absolute number of seeds produced in each fruit, instead of calculating the seed production relative to the ovules on each fruit.

We also measured seed germinability and seed viability. We haphazardly chose up to 20 seeds in each fruit and distributed them on filter paper in petri plates. Seed number was very low in the autogamy treatments; thus, seeds from spontaneous autogamy (Sa)

and facilitated autogamy (Fa) treatments were grouped. In total, we analyzed viability and germination rate of 1,800 seeds (10 plates per treatment and between 30 and 50 seeds per plate). Germination was induced with gibberellic acid 100 mg/L at 24 h of darkness and constant temperature of 23 °C (*Bell et al., 1995*). The state of germination and wetting of the plates were controlled on alternate days; the filter paper was removed every 2–3 days to reduce fungal infection. Seeds with a radicle of at least 2 mm were considered germinated. Seed germinability is the percentage of germinated seeds. We measured germination rate for a period of 45 days. After 45 days we tested seed viability by crushing the seeds with the tip of the tweezers. Soft and dark seeds were not considered viable and hard and light-colored seeds were considered viable (as *Hesse, Rees & Müller-Schärer, 2007*). These seeds along with the germinated seeds were considered the total number of viable seeds.

We calculated the Self-Compatibility Index (SCI) to describe the *G. lutea* breeding system (as *Lloyd & Schoen, 1992*). SCI is assessed as the average seed set for facilitated autogamy (Fa) divided by the average seed set for facilitated xenogamy (Fxw or Fxa) and gives information about the self-compatibility of the species. SCI values range from 0 to 1.5 and a species is considered self-incompatible when its values are between 0 and 0.75 (*Lloyd & Schoen, 1992*).

To assess the occurrence of self-fertilization, we calculated the Auto-Fertility Index (AFI) dividing the seed set for spontaneous autogamy (Sa) by the seed set for facilitated xenogamy within varieties (Fxw or Fxa) (*Lloyd & Schoen, 1992*). AFI gives information about the autonomous autogamy degree of the species.

In order to analyze the differences in seed number and seed weight among treatments, we used a generalized linear mixed model (GzLMM); the fixed factors were the treatment (five categories: C "Control group," Sa "Spontaneous autogamy," Fa "Facilitated auto-gamy," Fxw "Facilitated xenogamy within varieties," Fxa "Facilitated xenogamy among varieties"), the maternal flower color (color variety) and the interaction between color variety and treatment. Plant individual was a random factor nested within maternal flower color (color variety).

To test for differences in seed viability and germinability depending on treatment, we used a generalized linear model (GzLM) for each response variable. Treatment was a fixed factor (with four categories, since the seeds of spontaneous and facilitated autogamy treatments were joint to have a sufficient sample). Note that first, we performed a generalized linear mixed model (GzLMM), where treatment was a fixed factor and plate number was a random factor nested within treatment, but it did not converge. Additionally, we could not test if the germination rate varied between colors or individuals because of insufficient sample size. Error distribution and link function were selected to minimize the $AIC_C$: number of seeds was fitted to a Poisson distribution and a logarithmic link function; whereas weight of seeds was fitted to a Linear distribution and identity link function. Both seed viability and seed germinability were adjusted to a Binomial distribution and probit link function. Analyses were performed with SPSS software (*IBM Corp. Released, 2011*).

**Table 1 Effect of different treatments of pollination on female reproductive success (number of seeds, seeds weight, seed viability and seed germinability).** We marked in bold the statistically significant factors ($P < 0.05$).

| Response Variable | N | Factors | S.E. | Wald Chi-Square | d.f. | P |
|---|---|---|---|---|---|---|
| Number | 130 | *Random effect* | | | | |
| | | Plant (Color morph) | 0.016 | | | |
| | | *Fixed effects* | | | | |
| | | **Treatment** | | 504.389 | 4 | **0.000** |
| | | Color morph | | 0.001 | 1 | 0.971 |
| | | **Color morph * Treatment** | | 23.222 | 4 | **0.000** |
| Weight (mg) | 108 | *Random effect* | | | | |
| | | Plant (Color morph) | 3.112 | | | |
| | | *Fixed effects* | | | | |
| | | **Treatment** | | 44.347 | 4 | **0.003** |
| | | Color morph | | 0.004 | 1 | 0.949 |
| | | Color morph ∗ Treatment | | 0.177 | 4 | 0.950 |
| Viability (%) | 1,800 | *Fixed effects* | | | | |
| | | **Treatment** | | 30.906 | 3 | **0.000** |
| Germinability (%) | 1,800 | *Fixed effects* | | | | |
| | | **Treatment** | | 32.966 | 3 | **0.000** |

**Notes.**

Number, seed number; Weight (mg), seed weight ($N =$ fruits); Viability (%), proportion of viable seeds; Germinability (%), proportion of germinated seeds ($N =$ seeds).

## RESULTS

The most successful reproductive mechanism for *Gentiana lutea* was found to be cross-pollination between individuals within the same color variety. Conversely, cross-pollination among flower color varieties reduced plant reproductive success. In addition, we found that *G. lutea* depends on pollinators to reproduce.

Number of seeds, seed weight and germinability were similar for within-color variety crossings as for among-color variety crossings (Table 1, Figs. 1 and 2). However, seeds from crossings within varieties present 45% higher viability than seeds from among-variety crossings (Fig. 2). Thus, seed viability decreases when pollen donor and receptor belong to different color varieties (Fig. 2). This fact may imply that seed viability decrease is an important component of the partial hybridization barrier between *G. lutea* color morphs.

Fruits under autogamy treatments produced fewer (a reduction of 93% in the number of seeds) and lighter (a reduction of 37% in seed weight) seeds than fruits from inter-individual crossings. In addition, the effect of treatments on seed weight was the same for both color varieties (no significant color variety * treatment interaction: $P > 0.05$; Table 1 and Fig. 1); although, we found that under natural pollination, orange-flowering individuals produced more seeds per fruit than yellow-flowering individuals (significant simple effect of flower color: $P < 0.001$; Table 1 and Fig. 1). On the contrary, within the spontaneous autogamy treatment, yellow-flowering individuals produced more seeds per fruit however both morphs produced very few seeds (a reduction of 93% in the number of

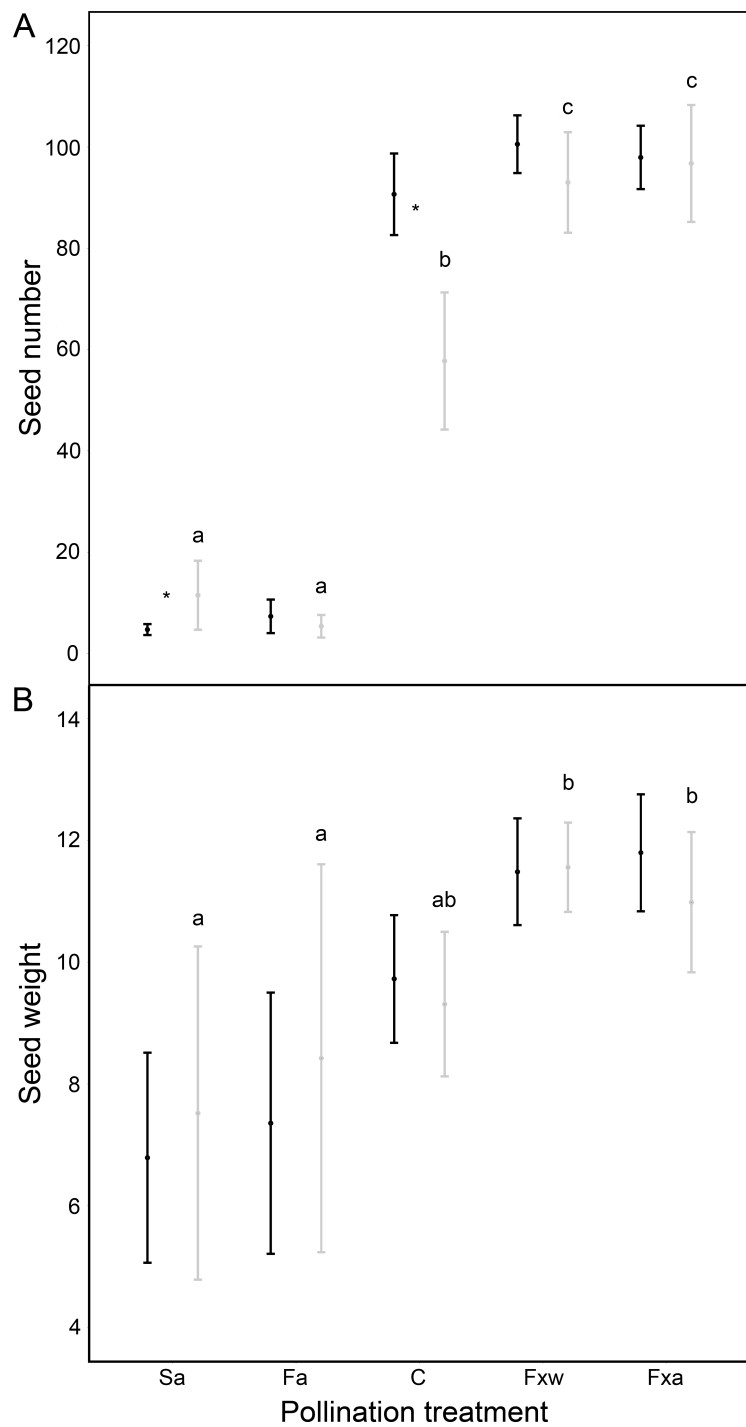

**Figure 1 Reproductive success assessed by seed number (A) and seed weight in mg (B), in function of different pollination treatments.** Pollination treatments were: Spontaneous autogamy (Sa); Facilitated autogamy (Fa); Control group (C); Facilitated xenogamy within varieties (Fxw) and Facilitated xenogamy among varieties (Fxa). Significant statistical differences between treatments ($P < 0.001$ for seed number and $P < 0.01$ for seed weight) are marked with different letters, and significant statistical differences between color varieties (orange: black-colored bars; yellow: grey-colored bars) are marked with an asterisk. Bars show the Standard Error (S.E.).
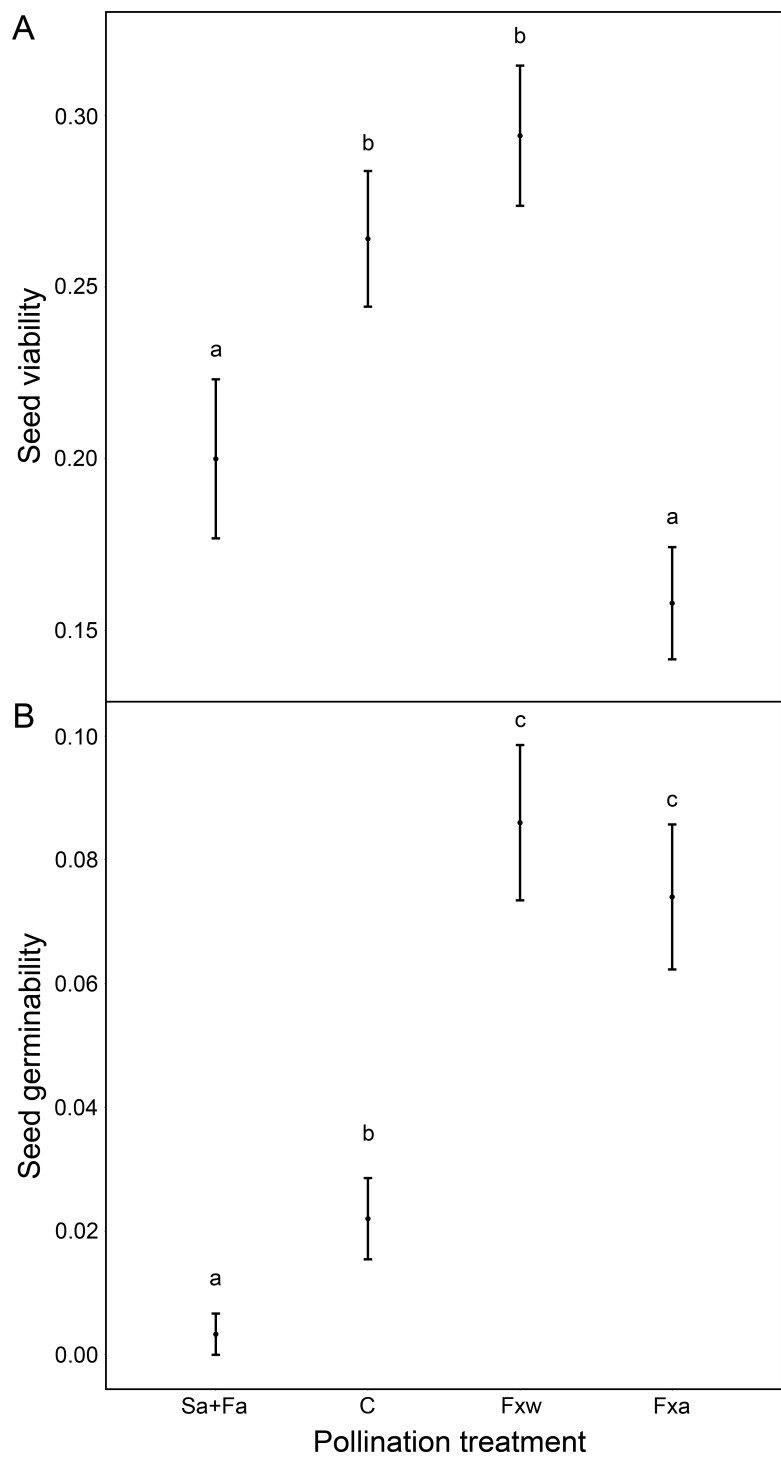

**Figure 2 Reproductive success assessed by seed viability (A) and seed germinability (B), in function of different pollination treatments.** Pollination treatments were the following: Spontaneous autogamy (Sa) + Facilitated autogamy (Fa); Control group (C); Facilitated xenogamy within varieties (Fxw) and Facilitated xenogamy among varieties (Fxa). The statistical significant differences between treatments ($P < 0.001$) are marked with different letters. Bars show the Standard Error (S.E.).

seeds) relative to outcrossed flowers ($P < 0.001$; Table 1 and Fig. 1). Additionally, we found significant differences among treatments for both, seed viability and seed germinability ($P < 0.001$; Table 1 and Fig. 2). Note that both xenogamy treatments resulted in greater seed germinability than the autogamy treatments and control group (Fig. 2).

The Self-compatibility index scored 0.0687. SCI ranges from 0 to 1.5, with values of SCI under 0.75 meaning that the species is self-incompatible. Thus, *Gentiana lutea* is a self-incompatible species which relies in cross-pollination to successfully reproduce. Auto-Fertilization Index (AFI) scored 0.0695 meaning that *G. lutea* is a non-autogamous species.

## DISCUSSION

The most advantageous reproductive mechanism for *Gentiana lutea* L. is cross-pollination between the same color morphs. Our results suggest that a partial hybridization barrier exists between *G. lutea* flower color morphs, likely due to a genetic incompatibility that reduces seed viability between *G. lutea* varieties. It seems that the differentiation between both varieties is driven or reinforced by the strong dependency of *G. lutea* on pollen vectors which exert selective pressures on flower color (*Veiga et al., 2015b*; *Sobral et al., 2015*).

Angiosperm diversification may derive from reproductive isolation due to changes in pollinator habitat composition (*Bradshaw & Schemske, 2003*; *Streisfeld & Kohn, 2007*; *Hoballah et al., 2007*; *Dauber et al., 2010*). A speciation process among flower color varieties of the same species can occur (*Straw, 1955*; *Waser, 1998*; *Gegear & Burns, 2007*) and may be driven by differences in pollinator community, which shows a higher preference for one morph as many studies suggest (*Dronamraju, 1960*; *Quattrocchio et al., 1999*; *Hopkins & Rausher, 2012*). We know that *Gentiana lutea*'s pollinators show different color preferences among populations at the Cantabrian Mountains (Spain), partially due to differences on the pollinator spectrum within each population and specific flower color preferences from each pollinator species (*Sobral et al., 2015*). Sympatric speciation could originate due to selective pressures exerted by the pollinators (*Quattrocchio et al., 1999*; *Hopkins & Rausher, 2011*) that facilitate isolation between flower color varieties when cross-pollinations are avoided or reduced (*Hopkins & Rausher, 2012*).

Additionally, the geographic isolation produced by the Quaternary climatic changes has been identified as the main cause of divergence in several mountain plant species (*Hewitt, 2000*; *Thompson, 2005*; *Gómez & Lunt, 2007*; *Vargas et al., 2009*; *Martín-Bravo et al., 2010*; *Alarcón et al., 2012*; *Blanco-Pastor & Vargas, 2013*; *Fernández-Mazuecos et al., 2013*). A subsequent secondary contact between both color morphs in *Gentiana lutea* L. could generate a similar situation to that shown in our study population and the surrounding area; whether it is a recent or an old contact, it seems that the maintenance of different color morphs is reinforced by the fitness reduction on hybrids and the pollinator behavior, as our results confirmed (Table 1, Figs. 1 and 2).

The Self-Compatibility Index obtained here suggests that *Gentiana lutea* L. is a self-incompatible species (*Lloyd & Schoen, 1992*). Therefore, this species relies on pollen vectors for a successful reproduction, as in the case of the most flowering-plant species (*Axelrod, 1960*; *Tepedino, 1979*). The Self-Compatibility Index (SCI = 0.068) results were

similar to the Auto-Fertilization Index (AFI = 0.069). Note that SCI is the ratio between the average seeds produced in facilitated autogamy treatment and the average seeds produced in facilitated xenogamy, whereas AFI is assessed from the ratio between seeds set from spontaneous autogamy and facilitated xenogamy treatments. Similar values of the two indexes imply that the number of seeds produced when a flower's own pollen arrives both, naturally and manually is similar. This fact may suggest that self-incompatibility in this species might be caused by some pre-zygotic barrier mechanism in which pollen from same flower is not fertilizing the ovules, or by some post-zygotic barrier that may produce a lower quality and number of self-pollinated seeds (*Charlesworth & Charlesworth, 1987*; *Hopkins, 2013*).

It is an interesting fact that seed number and weight, and to a lesser extent seed germinability, from natural pollination is intermediate between autogamy treatments and xenogamy crossings (see Figs. 1 and 2). This result suggests that pollen vectors carry to a plant, in natural conditions, both pollen from that plant and pollen from different plants. Therefore, those animal pollinators that have higher mobility between plants and lower mobility among flowers within plants would pollinate more successfully (*Dauber et al., 2010*; *Rossi et al., 2014*).

On the other hand, yellow-flowering plants set a greater number of seeds (than orange-flowering plants) when spontaneous self-pollination occurs, which might suggest that the self-incompatibility degree in this species varies among varieties and is actually higher for orange-flowering individuals (Fig. 1). Additionally, we found differences in seed number within the natural pollination treatment, in which orange flowers set a greater number of seeds. We know that pollinator assemblage shows preferences for yellowness within our study population, in which yellow-flowering individuals have a greater total seed set (*Veiga et al., 2015b*). However, our results suggest that although orange-flowering plants have lower fitness than yellow ones, they produce more seeds per fruit (see Table 1 and Fig. 1).

We know that pollinators exert selective pressures on *Gentiana lutea* L. flower color (*Veiga et al., 2015b*) and that these selective pressures drive flower color differentiation in this species (*Sobral et al., 2015*). Additionally, we know that abiotic factors such as temperature, radiation, elevation and rainfall are not related to flower color variation among *G. lutea* populations (*Veiga et al., 2015a*). With the available information, it is not clear whether color differentiation is due to an allopatric or sympatric process; however, our results bring to light the existence of a hybridization barrier among *G. lutea* color varieties.

## ACKNOWLEDGEMENTS

The authors thank reviewers for their comments and suggestions, P Domínguez for field assistance, L Salaverri for graphical edition and I Neylan for reviewing the manuscript.

### Funding

This study is included in the project "Color polymorphism, geographic variation in the interactions and phenotypic selection. The case of *Gentiana lutea* L. in the Cantabrian

Mountains" was financially supported by Secretary of State of I+D+I, Ministry of Science and Innovation, Government of Spain (2011–2013). The funders had no role in study design, data collection and analysis, decision to publish, or preparation of the manuscript.

## Grant Disclosures

The following grant information was disclosed by the authors:
Secretary of State of I+D+I Ministry of Science and Innovation, Government of Spain (2011–2013).

## Competing Interests

The authors declare there are no competing interests.

## Author Contributions

- María Losada performed the experiments, analyzed the data, wrote the paper, prepared figures and/or tables.
- Tania Veiga conceived and designed the experiments, performed the experiments, reviewed drafts of the paper.
- Javier Guitián, José Guitián and Pablo Guitián contributed reagents/materials/analysis tools, reviewed drafts of the paper.
- Mar Sobral analyzed the data, wrote the paper, prepared figures and/or tables.

## Field Study Permissions

The following information was supplied relating to field study approvals (i.e., approving body and any reference numbers):

Institution: Environmental Territorial Service from León, Regional Government of Castilla and León, Territorial Delegation of Government of Spain—Identifier: 12_LE_325_RNA_PuebladeLilio_INV—Reference: 06.01.013.016/ROT/abp—File number: AEN/LE/103/12.

## Supplemental Information

Supplemental information for this article can be found online at http://dx.doi.org/10.7717/peerj.1308#supplemental-information.

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
