# Peer review of "Is there a hybridization barrier between Gentiana lutea color morphs?"

_PeerJ, doi:10.7717/peerj.1308_

## Round 0.1 · original submission · Major Revisions

Both referees provided excellent feedback and suggestions. Please address as best you can.

·

Basic reporting

This article is well written, and in clear english. However, there are a few instances where word choice or sentence structure cause some confusion. Examples include: Line 151 "we controlled germination", do you mean you measured germination rate, or that this was a controlled study? Line 153 "viable and harsh", harsh doesn't make sense in this context. Lines 191 and 192 "were similar for within-color variety crossings than for among color..", 'than' should be 'as'. Line 220 "flower sap", is that different than nectar?

In the figure legends, the letters above the data points, indicating significance, should be clearly explained. Are they all the same level of significance? If not, how are they different?

Experimental design

no comments

Validity of the findings

no comments

Additional comments

The findings of this set of experiment are fairly straight forward and well defined. Your hypothesis was clearly stated and the experiments were correctly chosen to address that hypothesis. Apart from the minor wording issues, this paper is well written and easily readable. The attached PDF should have highlighting for the sentences that caused me some confusion.

Reviewer 2 ·

Basic reporting

The writing could be improved throughout, particularly the introduction and discussion where the text is incoherent in places, and poorly focused. See my comments to the authors below.

Experimental design

Experimental design and statistics are good!

Validity of the findings

No comments

Additional comments

This study examines mechanisms of reproductive isolation between sympatric flower-color morphs of Genetiana lutea in NW Spain. The primary goals were to determine the degree of post-pollination, prezygotic and intrinsic postzygotic isolation, and the degree to which the two morphs are dependent on pollinators for successful reproduction. The authors found an impressive 40-50% reduction in seed viability in hybrid crosses, and that both morphs are nearly completely dependent on pollinators for successful reproduction. Together, these results have interesting implications for reproductive isolation and co-existence of these two sympatric color-morphs, and set the stage for future work. In particular, the nearly 50% reduction in hybrid seed viability should strongly select for traits that reduce pollen movement between color morphs (e.g., phenological shifts to avoid co-flowering, or traits that increase pollinator constancy within color morph). Additionally, the strong correlation between flower color and seed viability begs the question of whether there is a major genetic incompatibility that is pleiotropic or just tightly linked with flower color.

The experimental design and analyses were nicely done. My primary advice is to tighten the introduction and discussion around the key findings, and to clarify some of the text. For example, after reading the first two paragraphs of the introduction, it seems like this paper is going to examine pollinator preference or constancy to flower color (paragraph 2’s main focus), which is not the case.

Line 15 and 16: This is not an either/or statement. It is possible for secondary contact and speciation to be underway.

Line 17 and 18: Do you mean here that flower color is heritable, and not determined by the local environment? Or do you mean that local abiotic environment does not select on flower color, as demonstrated by common gardens? Either way, although important to include in the main text, this seems an aside at this early stage in the abstract. Consider leaving this out of the abstract.

Line 18-19: The sentence “But, pollinators cause…” is not clear and has some grammar problems. Revise.

Line 21: Consider adding the word sympatric here, e.g., “Within a sympatric population containing both flower-color morphs, we…”

Line 24-25: Would help to state the magnitude of reduced seed viability, as it is quite impressive judging from Figure 2!

Line 26 and 27: Consider dropping the “Regardless of the sympatric origin of G. luetea flower color,”

Line 28-29: The conclusion that the hybrid incompatibility is “driven by pollinators” is misleading. The incompatibility resulting in reduced seed viability is likely a genetic incompatibility, but is not itself driven by pollinators. IF pollinators move between the two color morphs (not tested in this study), then this incompatibility will lead to substantially reduced fitness, contributing to reproductive isolation between the two color morphs.

Line 77-79: The last part of this sentence is unclear and speculative since it isn’t tested in this study: “, which might be caused or reinforced by selective pressures exerted by pollinators on G. lutea flower color.” Consider leaving this out.

Line 82-83: what do you mean by “under a diversification process”?

Line 134: Change to “proportion of viable seeds”

Line 135-136: This is not clear. Did you end up with 26 treatment plants total (down from the original 50)? How many treatment plants per morph then, 13 and 13?

Line 153: typo. Change “harsh” to “hard”.

Line 194-195: What is the percent decrease in seed viability for between relative to within morph crosses? It looks impressive from figure 2!

Line 197: Again, what is the percent decrease in the number of seeds in selfed vs. outcrossed fruits?

Line 203-204: You might stress here, that even though there was a significant difference between the two morphs in the number of automatically selfed seeds per fruit, both morphs produced very few seeds automatically relative to outcrossed flowers.

Line 216: The second half of this sentence is unclear: “,likely due to a past or ongoing divergence between G. lutea varieties.” Maybe instead you mean something like, “likely due to a genetic incompatibility that reduces seed viability”?

Line 287-289: This final sentence is more or less repeated several times earlier in the paper. It struck me as strange each time. Why are you emphasizing the “Regardless of the origin”? Also, do you mean the geographic origin being allopatric or sympatric? Also, because you don’t test (in this paper) whether pollinators are selecting on flower color, it seems strange to emphasize this point so many times. Avoid repetition and clarify or drop this sentence.

---

## Round 0.2 · accepted · Accept

Thank you for addressing the recommendations of both referees.